# Male Sexual Preference for Female Swimming Activity in the Guppy (*Poecilia reticulata*)

**DOI:** 10.3390/biology10020147

**Published:** 2021-02-12

**Authors:** David Bierbach, Ronja Wenchel, Stefan Gehrig, Serafina Wersing, Olivia L. O’Connor, Jens Krause

**Affiliations:** 1Department of Biology and Ecology of Fishes, Leibniz Institute of Freshwater Ecology and Inland Fisheries, Müggelseedamm 310, 12587 Berlin, Germany; ronjaw@zedat.fu-berlin.de (R.W.); stefan-gehrig@t-online.de (S.G.); serafina.wersing@igb-berlin.de (S.W.); j.krause@igb-berlin.de (J.K.); 2Faculty of Life Sciences, Humboldt-Universität zu Berlin, Invalidenstrasse 42, 10115 Berlin, Germany; olivia.oconnor@hu-berlin.de; 3Cluster of Excellence, Science of Intelligence (SCIoI), Technische Universität Berlin, Marchstraße 23, 10587 Berlin, Germany

**Keywords:** animal personality, swimming activity, male mate choice, mating preferences, *Poecilia reticulata*

## Abstract

**Simple Summary:**

In our first experiment, we found that male and female guppies differ consistently in their swimming activity, both when alone and when together in heterosexual pairs, and that males, regardless of their own activity, show more sexual behaviours to more active females in these pairs. In the second experiment, we gave males the choice between a slow and a fast-moving virtual female and found that males prefer to associate with the fast-moving female. Both experiments together provide evidence that guppy males prefer faster moving females as mating partners over slower moving ones. Higher activity patterns may indicate health, fitness and receptivity and a mating preference for more active females can help males increase their own reproductive success.

**Abstract:**

Mate choice that is based on behavioural traits is a common feature in the animal kingdom. Using the Trinidadian guppy, a species with mutual mate choice, we investigated whether males use female swimming activity—a behavioural trait known to differ consistently among individuals in many species—as a trait relevant for their mate choice. In the first experiment, we assessed male and female activity in an open field test alone (two repeated measures) and afterwards in heterosexual pairs (two repeated measures). In these pairs, we simultaneously assessed males’ mating efforts by counting the number of sexual behaviours (courtship displays and copulations). Male and female guppies showed consistent individual differences in their swimming activity when tested both alone and in a pair, and these differences were maintained across both test situations. When controlling for male swimming behaviour and both male and female body size, males performed more courtship displays towards females with higher swimming activity. In a second experiment, we tested for a directional male preference for swimming activity by presenting males video animations of low- and high-active females in a dichotomous choice test. In congruence with experiment 1, we found males to spend significantly more time in association with the high-active female stimulus. Both experiments thus point towards a directional male preference for higher activity levels in females. We discuss the adaptive significance of this preference as activity patterns might indicate individual female quality, health or reproductive state while, mechanistically, females that are more active might be more detectable to males as well.

## 1. Introduction

Males, the same as females, can exercise mate choice [1,2]. Male mate choice is not only evident in species with reversed sex roles and male parental care [3,4], but also in species where male investment is low but females differ in quality [5,6,7]. Female quality in such species is often approximated by fecundity, and, as a correlate, female body size [2,8]. In addition, males also choose their mating partners on the basis of coloration [9], actual mating status [10] or quality of parental care [11]. However, some studies on female mate choice report on sexual preferences for behaviours not directly related to reproduction (see [12,13]) such as feeding rates [14], exploration tendencies [15,16] or predator inspection behaviour [17].

Similarly, males may choose females based on nonsexually motivated behaviours if these behaviours are consistent indicators of quality (see [13,18,19]). Alternatively, males may benefit when mating with females that match their own behavioural phenotypes [13,20]. In zebra finches, females prefer males that exhibit similar exploration tendencies to themselves [15] and pairs with similar behavioural types have increased parental care success [21]. In guppies, females paired with males who differed from them in risk-taking behaviour had a lower parturition success than females who mated with males with a similar degree of risk-taking [22].

In the current study, we ask whether males use female activity—a behavioural trait known to differ consistently among individuals in many species [23,24,25]—as a criterion in mate choice. Activity patterns can indicate female quality as they often positively correlate with metabolism [26,27,28], growth rate [29,30], general health [31], as well as survival [18,19,32]. In addition, activity patterns may even indicate reproductive status, especially in livebearers where females carry developing embryos and eggs and thus may lose movement agility when pregnant or gravid [33,34]. Mechanistically, more active females might be more conspicuous to males as found for prey that is more visible to predators when moving more (see [35,36]).

We decided to use the live-bearing Trinidadian guppy (*Poecilia reticulata*) as a model organism as both males and females are known to exercise mate choice [37] and individuals often differ consistently in their activity levels (with males being on average more active than females [38,39]). In this species, higher activity levels in females correlate positively with risk-taking behaviours [40], and negatively with gestation period [34] and body size [40].

We hypothesized that males should show a general preference for more active females (“directional preference hypothesis”) as found, for example, in lizards [16]. As an alternative hypothesis, we explored whether males prefer females with an activity type that is similar to their own (“phenotypic matching hypothesis”), as found in zebra finches [21]. To provide evidence for either hypothesis, in the first experiment we assessed male and female activity in an open field test alone (nonsexual context treatment, two repeated measures) and in heterosexual pairs (sexual context treatment, two repeated measures). In these pairs, we simultaneously assessed males’ mating efforts by counting the number of sexual behaviours (courtship displays and copulations). Sexual behaviours exhibited by male guppies are a direct measure of sexual preference [5,10,41]. For the directional preference hypothesis, we predicted that males should show the largest numbers of sexual behaviours towards the most active females, while for the phenotypic matching hypothesis, males should direct most sexual behaviours to females with activity levels similar to their own.

As the first experiment will provide purely correlational evidence, we performed a second experiment specifically testing the directional preference hypothesis using a dichotomous choice design. Males could choose to associate with computer-animated females that were either fast or slow swimming. Here, the prediction was that males would spend more time close to the faster animated females.

## 2. Materials and Methods

### 2.1. Study Organism

Test fish were laboratory-reared descendants of wild-caught Trinidadian guppies that originate from the Arima river in Trinidad (low predation site; see [37]). They came from large, randomly outbred, mixed-sex stocks reared under constant physical conditions (25 °C, 12 h of light). The fish were fed three times per day with commercially available flake food (TetraMin^®^ by Tetra GmbH, Melle, Germany); water was exchanged once a week. Prior to the experiments, males and females were separated by sex for one week to increase sexual motivation [5]. After the experiment, all fish went back into their stock tanks and did not serve in any other test. 

### 2.2. First Experiment

All tests were conducted in a standard tank (35 cm × 20 cm) that was filled with water from the holding tanks to a level of 5 cm. The walls were covered to avoid disturbances from the outside and observations were carried out via a webcam (Logitech 920c by Logitech Inc., Appels, Switzerland ) fixed at 50 cm above the tank. Male activity was measured by introducing a male into a Plexiglas cylinder inside the test tank. After a habituation of 5 min, the cylinder was lifted and the activity of the test fish was videotaped for another five minutes. This testing was repeated for all individuals after 48 h (Figure 1A). After the first test, males were housed separately in small tanks of a circulating water system. This system contained, besides males also females and all fish thus had experienced both visual as well as olfactory cues from other, same and heterosexual conspecifics. 

Female activity was measured as described for males but each 5 min activity test was followed by another phase in which we presented females with a male (Figure 1). To do so, females were placed back into their cylinders. Then, a randomly selected male from our pool of already tested males was introduced into the test tank. After five minutes, the cylinder was removed and both fish were videotaped for five minutes. As for the male activity test, this two-phase test was repeated after 48 h with another male. 

From the videos, we extracted the total distance moved by each fish during the five minutes tested alone and in dyads as a proxy for individual activity [42]. In the dyads, we further counted the number of copulation attempts as well as courtship displays by the males directed to the females as a proxy for male sexual effort. During copulation attempts, males swing their intromittent organ, the gonopodium, forward while attempting to introduce it into the female’s gonopore. Courtship in guppies is characterized by males courting in front of females in an S-shaped body posture (see [5] for details). To extract the distance, videos were analysed using the tracking software EthoVision 10 XT (Noldus Information Technology Inc., Wageningen, the Netherlands) with 15 frames per second and a smoothing profile of 2 mm minimum distance moved (MDM) based on five samples before and after every sample point. The standard length (head to base of tail) of each fish was measured to the nearest millimetre via photos using ImageJ software (ImageJ 1.46a by Fiji). In total, 82 females (standard length (SL): 18.78 ± 0.37 mm, mean ± standard error (SEM)) and 52 males (SL: 13.81 ± 0.31 mm) were tested and their size range is comparable to natural observations [43]. 

We first assessed whether males and females showed consistent individual differences in their activity levels when tested alone (nonsexual context treatment). To do so, we calculated the behavioural repeatability with linear mixed models (LMMs) following the protocol by Dingemanse and Dochtermann [44]. We included “trial” as a fixed factor to account for average differences in activity between first and second personality assessments (Figure 1) and individual “body size” (SL) as a covariate to evaluate whether activity correlates with body size. The fish ID was specified as a random factor to allow the decomposition of the variance into within- and among-individual components and subsequent calculation of behavioural repeatability. This “conditional repeatability” is a measure for consistent differences among individuals and its *p*-values derive from likelihood ratio tests (LRTs) against χ^2^ distributions with 1 degree of freedom [45]. 

Then, we asked whether females also differ consistently in their activity when together with a male (sexual context). Thus, we calculated the repeatability of female activity in dyads using a similar LMM as described above. As swimming performance is known to be influenced by social partners [46], we included both activity of the male partner and its body size as covariates in the model. To compare average activity levels across treatments, we ran an LMM with “trial” and “treatment” (indicating sexual or nonsexual contexts) as well as their interaction as fixed factors. Female ID was used as random factor. 

To assess whether females differed consistently across treatments (alone and in a pair), we calculated the among-individual correlation of female activity using a bivariate LMM. To do so, we first standardized female activity within both treatments and measures to a mean of 0 and a SD of 1 (*z*-transformation). Thus, no average differences between the treatments and measures may affect the among-individual correlation. In a second step, activity levels of females alone and in the dyads were used as the dependent variables with Female ID as a random factor. We allowed within-individual as well as among-individual variances to differ between treatments as well as to correlate within treatments (see [40,47] for similar approaches). As for repeatability, LRT tests were employed to assess whether the inclusion of within- and among-individual correlations significantly improved the model.

Our main question was whether males directed more sexual effort towards females who swam more actively (directional preference) or whether male sexual effort depended on both male and female activity levels (phenotypic matching). We thus used “received male sexual efforts” (sum of copulation attempts + courtship displays per pair) as the dependent variable in a generalized linear mixed model (GLMM) with Poisson error-distribution and Log-link function as count data. As fixed factors, we included “trial”, “female activity” (within the dyad), “male activity” (within the dyad; residuals from linear regression with female activity as both variables were highly correlated, see Results) as well as male and female absolute body size (“SL female”; “SL male”). We further included the interaction between “female activity” (within the dyad) and “male activity” (within the dyad). Female ID was used as a random factor. A significant effect of the term “female activity” would indicate a directional male preference, while a significant effect of “male activity” would indicate that male activity determined their sexual efforts. If the terms “female activity” and “male activity” were significant and positive, more active males would have directed more sexual efforts to more active females. A significant interaction term of both effects would indicate a mismatching sexual preference as less active males would direct more sexual efforts to high-active females. However, we removed the interaction term from the final model as there was little evidence supporting that it was different from zero in either direction (see Results). 

### 2.3. Second Experiment

Using a dichotomous choice design (see [48]), we evaluated male preferences (new set of males, total length (TL ± SEM): 20.4 ± 0.5 mm; *n* = 17) for two virtual stimulus female types (sized on screen set to 30.0 mm TL each) by showing each focal male video animations of the same virtual female either swimming at 0.7 cm/s or with 2.7 cm/s and measured association preferences for 10 min (with switched side assignment after 5 min). The used swimming speeds were the lower and upper 10% percentiles of females from experiment 1. This technique is well established in the evaluation of mating preferences in fish per se [49,50,51,52,53] and has been used successfully by the authors in previous studies [48,54].

To produce the video animations, digital images of 5 females were taken with grey background colour using a Canon 400D (Canon Inc., Tokio, Japan) digital camera (see Appendix A). From each resulting picture, the image of the female was extracted from the background using the “magic wand” selection tool in Photoshop CS5 (Adobe Inc., San Jose, USA). Each picture was then embedded into a Microsoft PowerPoint presentation and a straight movement of the picture from left to right and right to left was generated in front of a uniformly light grey background. For each picture of a female, we generated a slow and a fast-moving version. Simultaneous playback was performed using two identical computer monitors (P2419H, Dell Inc., Round Rock, USA) as well as two laptops with identical MS Office versions (Office 365+, Microsoft Inc. Redmond, USA). The monitors’ refresh rates were 60 Hz and a sample animation (.mp4) is available as a supplement (Appendix A).

For the preference tests, monitors were placed on either side of a test tank (40 cm length × 25 cm width × 25 cm depth) that was visually divided into three sections: two preference zones (5 cm length) adjacent to the monitors and a central neutral zone (30 cm length). Both long sides of the tank were covered by white Perspex, so that the focal male did not see the experimenter in the room. Water level was kept at 15 cm, which was also the height of the monitors. Water temperature was maintained at 25 °C, and illumination was provided through two LED stripes on the ceiling of the experimental room. The focal male was observed via a acA1920-155um USB3 camera (Basler AG., Ahrensburg, Germany) in full HD with 30 fps and the resulting videos were analysed using Ethovision software.

To initiate a trial, we introduced a male into the test tank. After a habituation period of 5 min (showing only a grey video screen), we started a 5 min observation period during which we measured association times, i.e., times spent in each preference zone, near the monitor showing the animation of the fast- or slow-moving female. To account for potential side biases, we interchanged the stimulus animations and repeated measurement of male preferences after another 5 min of habituation. We decided a priori to assume a side preference when a male spent 80% of its association time during both test units in one of the two preference zones (i.e., at only one side of the test tank, irrespective of the stimulus animation). Based on this criterion, 6 out of 23 trials had to be discarded from the final analysis, resulting in a final sample size of 17 males. We summed the times spent near either female animation during the two test units and further extracted male swimming velocity and body size from the videos. Association preferences are well established to translate into reproductive outcome in poeciliid fishes [55]

We first compared the time males spent near each stimulus using paired-samples t-tests. In the next step, we wanted to see how male body size, male swimming activity as well as stimulus females’ picture IDs may have affected male mate choice. To do so, we calculated a strength of preference (SOP) score as the time spent near the fast-swimming animation minus the time spent near the slow-swimming animation, divided by the total time spent choosing. Thus, a SOP score of +1 reflects the maximal preference for the fast stimulus animation and −1 maximal preference for the slow stimulus animation. SOP scores were then used as dependent variable in an LMM with male body size, male swimming activity as covariates as well as stimulus females’ picture IDs as a random factor. An additional LMM on SOP score was fit as a robustness check for the paired t-test on raw time spent near stimulus mentioned above. Since this LMM contained no fixed effects, the intercept was an estimate for the marginal strength of preference for females, adjusted for stimulus females’ picture IDs. All data analyses were conducted in SPSS version 25 (IBM Inc., Armonk, USA).

## 3. Results

### 3.1. First Experiment

We found significant among-individual variation in male and female activity when fish were observed alone (repeatability_males_ = 0.76, *p* < 0.001; repeatability_females_ = 0.63, *p* < 0.001). On average, activity significantly decreased with repeated testing in both sexes (males: *F*_1,52_ = 78.06, *p* < 0.001; females: *F*_1,81_ = 67.60, *p* < 0.001). In females, activity negatively and significantly covaried with body size (*F*_1,80_ = 4.81, *p* = 0.031), meaning that larger females were less active (*posthoc* Pearson’s correlation: *r*_p_ = −0.201; *p* = 0.010).

Together with males in a pair, females showed consistent among-individual variation (repeatability = 0.45, *p* < 0.001) and average female activity covaried positively and significantly with the male partner’s activity (*F*_1,149_ = 230.85, *p* < 0.001; *posthoc* Pearson’s correlation: *r*_p_ = 0.80; *p* < 0.001). There was some evidence that females, on average, decreased their activity with repeated testing (*F*_1,74_ = 3.7, *p* = 0.057). However, there was no significant effect of female or male body size on female activity levels (female body size: *F*_1,81_ = 0.39, *p* = 0.845; male body size: *F*_1,149_ = 2.51, *p =* 0.115).

When comparing average female activity among treatments, we found females to be less active when tested with a male than when tested alone (Figure 2A; treatment: *F*_1,243_ = 276.20, *p* < 0.001). Similar to the model results above, females were found to reduce their activity with repeated testing when pooling data from both treatments (Figure 2A; trial: *F*_1,243_ = 61.80, *p* < 0.001). However, the decrease in activity from a nonsexual to sexual context treatment was more pronounced in the first personality assessment (Figure 2A; “treatment × trial”: *F*_1,243_ = 11.92, *p* = 0.001). Despite these differences, we found strong support for among-individual correlation in activity across treatments (*r*_among_ = 0.73, *p* < 0.001, Figure 2B). There was also a less pronounced within-individual correlation (*r*_within_ = 0.25, *p* = 0.021). Thus, females that were on average more active alone were also more active when tested together with a male (Figure 2B, among-individual correlation). Furthermore, females that were more variable between repeated trials in their activity alone were also more variable in activity when with a male (significant within-individual correlation).

Our GLMM with the received sexual efforts as the dependent variable found a strong positive relationship between male sexual efforts and female activity (“female activity (dyad)”: *F*_1,157_ = 9.51, *p* = 0.002). This suggests that more active swimming females received more sexual behaviours than less active ones (Figure 3). None of the other predictors had a significant effect (“male activity” (residualised): *F*_1,157_ = 0.19, *p* = 0.661; “female SL”: *F*_1,157_ = 0.31, *p* = 0.578; “male SL”: *F*_1,157_ = 1.57, *p* = 0.21; “trial”: *F*_1,157_ = 2.60, *p* = 0.109; interaction “female activity by male activity” (not in the final model): *F*_1,156_ = 1.20, *p* = 0.275). Thus, there was no evidence that male sexual effort was related to body size of the involved fish, males’ swimming activity or to the interaction of male and female activity.

### 3.2. Second Experiment

We found that males spent significantly more time near fast-moving female animations as compared to slow-moving ones (*t*_16_ = 2.209; *p* = 0.0421, Figure 4). The strength of these preferences (SOP score) appeared to be independent of male body size (LMM: *F*_1,14_ = 0.08; *p* = 0.79) as well as swimming speed during the mate choice trials (*F*_1,14_ = 0.94; *p* = 0.35). Analysing SOP scores in an LMM without any fixed effects (but a random effect for stimulus females’ picture ID) confirmed the overall preference for fast-moving females. The model intercept was positive and significantly different from zero (estimate: 0.226, SE: 0.100, *t*_16_ = 2.263, *p* = 0.038).

## 4. Discussion

We found that male and female guppies show consistent individual differences in their swimming activity when tested alone as well as in heterosexual pairs. In support of the directional preference hypothesis, more actively swimming females received higher numbers of sexual behaviours by males, regardless of female or male body size or male swimming activity. Our second experiment confirmed the correlational evidence for a male preference for female swimming activity from experiment 1 by showing that males spent more time near fast-moving female animations compared to slow-moving ones in dichotomous mate choice tests.

Levels of individual swimming activity in an open field arena were found to be consistently different among individuals (i.e., there was considerable and significant repeatability) when males and females were tested both alone and together (=significant among-individual correlation). These results are consistent with previous studies showing significant repeatability in activity levels in the guppy [38,56,57,58] even when interacting with (artificial) conspecifics (see [40,59,60]). Those individual differences in swimming patterns can be maintained even in larger groups as shown for mosquitofish (*Gambusia holbrooki*, [46]) and sticklebacks, *Gasterosteus aculeatus* [61]. Thus, swimming activity fulfils a prerequisite of a trait that could be used as a cue in mate choice [62,63]. For male mate choice, we assume that activity patterns can indicate female quality as it often positively correlates with metabolism, growth rate, general health, as well as survival (see Introduction). On the other hand, the fact that more active individuals are also more visible to predators [35] may render high activity levels a costly trait that could fulfil the requirements of the “handicap principle” (after [64]). Following the same logic and in addition to the aforementioned indication of quality, higher activity levels may increase an individual’s visibility towards the opposite sex or even stimulate more exaggerated sexual responses (“arousal hypothesis”). It is further known that gestation statuses in live-bearing animals where developing eggs or embryos are carried by females hamper the female’s agility and mobility [33,34]. Female guppies are most receptive a few days postpartum [65] and males typically need to assess females’ reproductive status through nipping at the female’s gonopore [66] or by observing and imitating the mate choice of other males (mate choice copying, [67]). Both are assumed to be costly for the males in terms of opportunity costs [68]. It is thus possible that male guppies use female activity patterns to select females that gave birth recently and are thus most receptive. Future studies are thus recommended to directly test for the several not mutually exclusive drivers of the herein found male preference.

As predicted by the “directional preference hypothesis”, males generally increased their sexual efforts when paired with females that were more actively swimming (experiment 1). Contrary to the “phenotypic matching hypothesis”, this preference was independent of male activity (and its interaction with female activity) as well as body size of males and females. Although guppy males can perform courtship prior to copulation attempts, their mating system is assumed to be coercive [37,69,70] and forced copulations and sexual harassment are common features both in the lab and in the wild [71,72,73,74]. Thus, a phenotypic matching of activity with potential reproductive benefits such as, as an example, synchrony during mating or egg/sperm release, may not play a significant role in the guppy compared to other species [21,75]. This view is further supported by our second experiment where males spent significantly more time in association with the more active female animation regardless of own size or swimming activity. Future work on postcopulatory effects of mating partners’ swimming activity and overall reproductive success is needed to investigate the possibility of such a pattern in more detail. Nevertheless, if female swimming activity indicated overall quality or reproductive status (as our experiments suggest), a mating preference that leads to phenotypic matching for activity in pairs might not be beneficial in this species.

Female activity was lower in the second trial than in the first one and lower in the sexual than in the nonsexual context treatment. Reducing swimming activity in the presence of males could be an adaptive strategy to avoid male harassment if females anticipate increased sexual efforts by males as a response towards high swimming activity. A study by Killen et al. [71] found, however, that females that experienced high male sexual harassment (many males in holding tank; sex ratio 1:1) were more active than females that experienced only low levels of harassment (few males in holding tank; sex ratio 1:5.7). Additionally, females from the high harassment treatment swam more efficiently (less oxygen needed at a given speed), spending less time performing costly pectoral fin-assisted swimming compared to low harassment females. In contrast to our study, these authors recorded activity in the holding tanks and their study lacked a treatment with no males present. We thus argue that the activity reduction in response to a male stimulus which we found could represent a short-term behavioural adaptation to male harassment. In the long run, increased male harassment could result in more efficient swimming performance by females due to experience [71] and exercise [76], ultimately driven by the opportunity costs associated with male harassment. The general reduction in swimming activity from trial one to trial two in our study is most likely an effect of habituation to the experimental setup and the test environment (see [77]).

Why was there no evidence for a preference for large female body size in our study? Similar experiments in the mosquitofish (*G. holbrooki*) revealed increased sexual activity towards larger females [41]. In the Trinidadian guppy, evidence for body size preferences in males is inconclusive. In a study by Ojanguren and Magurran [10], it was found that males of one guppy population show such a preference (lower Aripo, high predation site), but not the males of another one (upper Aripo, low predation site). It thus seems that male preferences for a large female body size in the guppy is a population-specific trait and may further depend on various ecological factors (see results of Kniel and Godin [78] as well as their discussion for further details). We therefore encourage future experimentation on population differences in male preferences for female activity and body size (see [79] for a review on variation in female preferences among populations).

## 5. Conclusions

In conclusion, our study provides first evidence for a male sexual preference for high female activity levels in the guppy. Activity differs consistently among females and may indicate both fitness and reproductive status in live-bearing animals. Increased activity levels further enhance visibility or arouse the mating partner. All this can be used to propose that a directional preference for higher activity levels is an adaptive strategy for males in guppies but also other species. Future work on male mating preferences for activity levels in other species as well as the postcopulatory effects of such preferences on reproductive fitness may well be recommended, in particular when assuming activity differences play a ubiquitous role in the animal kingdom. 

## Figures and Tables

**Figure 1 biology-10-00147-f001:**
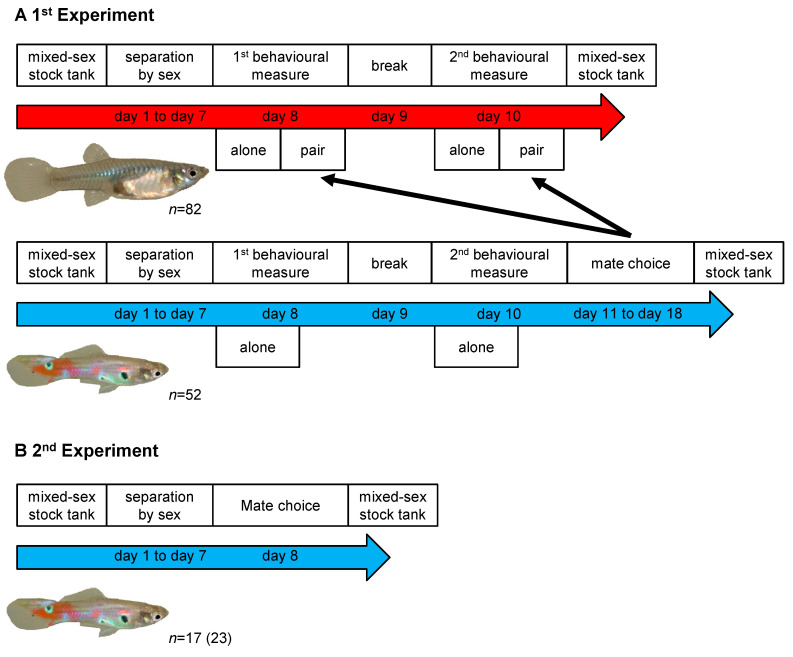
Schematic overview of the experimental timeline for experiments 1 (**A**) and 2 (**B**).

**Figure 2 biology-10-00147-f002:**
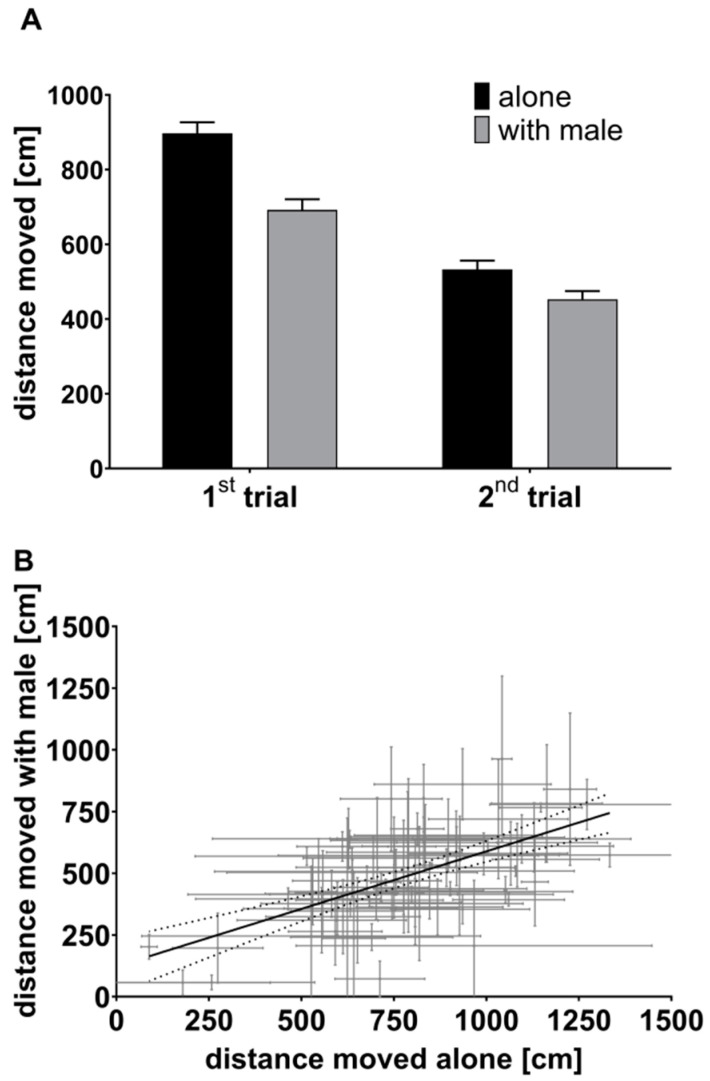
(**A**) Distance moved by females. Mean distances for both trials (1st and 2nd) are shown as well as for the treatments (female alone or with a male partner). (**B**) Consistency in female swimming activity across treatments. A scatter plot with individual-level error bars and a linear line of best fit (with 95% CI, dotted lines) is shown for female distances moved in an open field arena both alone and with a male partner. Error bars represent standard error (SEM) (**A**) and SD (**B**).

**Figure 3 biology-10-00147-f003:**
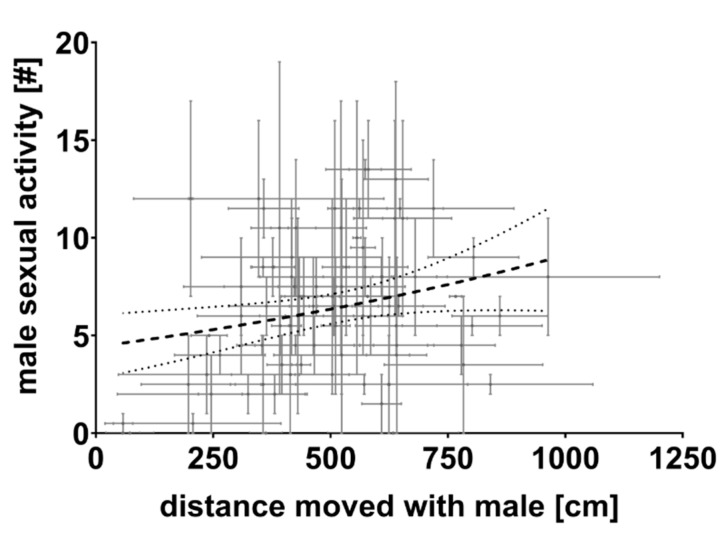
Male sexual activity (sum of courtship displays and copulation attempts) and female swimming activity (distance moved in an open field tank with a male partner). The regression line refers to the final generalized linear mixed model (GLMM) predictions (with 95% CI). Error bars represent SD.

**Figure 4 biology-10-00147-f004:**
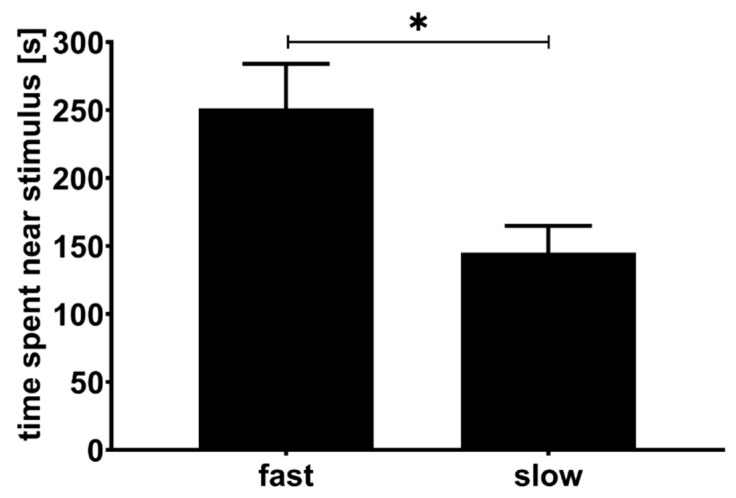
Results of dichotomous mate choice tests to assess directional mating preferences of male guppies for female swimming activity. The mean (±SE) times males spent near fast (2.7 cm/s) and slow (0.7 cm/s) moving female animations are shown. An asterisk indicates significant difference (*p* < 0.05) in paired-samples *t*-test.

## Data Availability

All data are available as supplemental material.

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
