# Peer review of "Male Sexual Preference for Female Swimming Activity in the Guppy (Poecilia reticulata)"

_biology, 2021, doi:10.3390/biology10020147_

Round 1

Reviewer 1 Report

The paper by Bierbach et al. investigates male sexual preference with respect to female swimming activity. They found that males prefer faster moving females as mating partners since higher activity patterns may indicate overall higher female quality and a mating preference for more active females could increase male reproductive success. I think that the topic of manuscript is quite interesting, especially since the shifting of the sexual selection paradigm of choosy females towards the male sex, although not completely novel, is still relatively overlooked. The manuscript is well-written and I found the experimental design sound, carefully explained and therefore easy to follow, and the results are clearly presented and discussed. Thus, I have only minor suggestions for the authors.

Simple summary: from the first sentence the reader can understand that males and females change their swimming activity when alone with respect to when in pair, but it is not clear if the authors are referring to male-female couples or also same-sex couples. It is stated immediately after in the abstract that this is referred to heterosexual pairs, but this very first sentence could be rephrased a bit to make it immediately clear from the start.

Introduction: when referring to species when males exert mate choice based on female quality please cite also these papers that investigate male choice on insect species based also on variance in female quality, sex ratio and female caste:

Kvarnemo, C., & Simmons, L. W. (1999). Variance in female quality, operational sex ratio and male mate choice in a bushcricket. Behavioral Ecology and Sociobiology, 45(3), 245-252.

Nandy, B., Joshi, A., Ali, Z. S., Sen, S., & Prasad, N. G. (2012). Degree of adaptive male mate choice is positively correlated with female quality variance. Scientific Reports, 2(1), 1-8.

Cappa, F., Bruschini, C., Cervo, R., Turillazzi, S., & Beani, L. (2013). Males do not like the working class: male sexual preference and recognition of functional castes in a primitively eusocial wasp. Animal Behaviour, 86(4), 801-810.

In paragraph 2.2 the authors refer to [5] for a description of courtship displays and copulation attempts. Maybe it would be possible to add also a short description of these behaviors as supplementary materials. In this way it will be more immediate for the reader.

page 4 Line 25: correct “und” with “and”.

In discussion the authors state that “more active individuals are also more visible to predators [35] may render high activity levels a costly trait that could fulfil the requirements of the handicap principle”; but it is also possible that more active individuals could be more responsive to outer stimuli and more able to escape potential predators compared to less active ones. The authors might discuss this possibility.

Since there are no line numbers it was a bit difficult to report mistakes, but I the text should be carefully checked to correct little spelling errors, punctuation and double or missing brackets.  

Reference 16: add journal pages.

Reference 20: check journal pages.

Reference 25: add journal pages.

Reference 30: add journal pages.

Reference 32: add journal pages.

Reference 36: check journal pages.

Reference 42: missing journal.

Reference 46: check journal pages.

Reference 47: add journal pages.

Reference 54: check journal pages.

Reference 68: check journal pages.

Reference 71: check journal pages.

Reference 72: check journal pages.

Reference 73: check journal pages.

Author Response

REVIEWER 1

Simple summary: from the first sentence the reader can understand that males and females change their swimming activity when alone with respect to when in pair, but it is not clear if the authors are referring to male-female couples or also same-sex couples. It is stated immediately after in the abstract that this is referred to heterosexual pairs, but this very first sentence could be rephrased a bit to make it immediately clear from the start.

Response:  Thank you very much for this suggestion, we changed the first sentence accordingly.

Introduction: when referring to species when males exert mate choice based on female quality please cite also these papers that investigate male choice on insect species based also on variance in female quality, sex ratio and female caste:

Kvarnemo, C., & Simmons, L. W. (1999). Variance in female quality, operational sex ratio and male mate choice in a bushcricket. Behavioral Ecology and Sociobiology, 45(3), 245-252.

Nandy, B., Joshi, A., Ali, Z. S., Sen, S., & Prasad, N. G. (2012). Degree of adaptive male mate choice is positively correlated with female quality variance. Scientific Reports, 2(1), 1-8.

Cappa, F., Bruschini, C., Cervo, R., Turillazzi, S., & Beani, L. (2013). Males do not like the working class: male sexual preference and recognition of functional castes in a primitively eusocial wasp. Animal Behaviour, 86(4), 801-810.

Response: Thank you very much for this suggestion. In fact we cite insect male mate choice by referring to Bonduriansky, R., The evolution of male mate choice in insects: a synthesis of ideas and evidence. Biological Reviews, 2001. 76(3): p. 305-339. Thus, we don’t feel that more, species-specific papers are necessary for the story here. However, we would leave the decision to include these papers with the editor.

In paragraph 2.2 the authors refer to [5] for a description of courtship displays and copulation attempts. Maybe it would be possible to add also a short description of these behaviors as supplementary materials. In this way it will be more immediate for the reader.

Response: This is a very nice suggestion and we added the following sentences to that paragraph: “During copulation attempts, males swing their intromittent organ, the gonopodium, forward while attempting to introduce it into the female’s gonopore. Courtship in the guppy is characterized by males court in front of females in an S-shaped body posture (see [5] for details).”

Page 4 Line 25: correct “und” with “and”.

Response: corrected

In discussion the authors state that “more active individuals are also more visible to predators [35] may render high activity levels a costly trait that could fulfil the requirements of the handicap principle”; but it is also possible that more active individuals could be more responsive to outer stimuli and more able to escape potential predators compared to less active ones. The authors might discuss this possibility.

Response: The reviewer is correct, and we bring up both points in our introduction and discussion. First we say that activity might correlate with survival (citing two meta-analysis as well as an empirical study using the guppy, references #18,19 and 32) but also pointing towards evidence that faster movements can increase visibility and detectability (ref. # 35). We now added a sentence stating that future studies are needed to test which of these hypotheses may explain the found effect best.

Since there are no line numbers it was a bit difficult to report mistakes, but I the text should be carefully checked to correct little spelling errors, punctuation and double or missing brackets. 

We are sorry for this! We used the journal’s word template which does not seem to support line numbers.

Various edits and corrections to the reference list were done.

Reviewer 2 Report

This study examines sexual preferences in male Trinidadian guppies, and whether female activity is an attractive trait. Overall the study is well designed and interesting, it provides some new context for this system, and raises some interesting questions worthy of further research. I generally have only minor suggestions, although (as detailed in my comments) I have some concerns over what the researchers actually measured.

In section 2.1: The authors state that males and females were separated by sex; was water mixed between these tanks in a system, or were all tanks completely separate?

1st experiment: Was there any particular reason why water from the holding tanks was used? Is there a possibility that cues from the holding tank were therefore maintained within the experimental tank which could have influenced the result? For example are males likely to behave differently (more actively) in the presence of cues from male conspecifics? The authors do indicate that fish had experienced olfactory cues of other conspecifics so perhaps this isn't important. Presumably however holding tanks were separate (rather than in a system sharing water; see my first point) and therefore the water used in the experiment did not contain cues from the alternate sex?

One concern I have with the methodology is that activity as measured here could be conflated with exploration i.e. examining a new environment rather than moving around per se. Have the authors considered whether it is indeed activity they have measured? This may be particularly true since activity appears to be lower in the second trial (and in the discussion attributed to habituation). I think at the least within the methods the authors should reference other studies that have been conducted in this way and refer to the measure taken as ‘activity’. It is really important in behavioural science to ensure that the terminology used accurately reflects the behaviours being assessed, or at least that researchers are using the same terminology across studies. The authors do refer to both activity and exploration in the introduction.

on page 4 paragraph 3 the authors refer to ‘trail’ but I think this is a typo of ‘trial’?

What was the fate of the fish? Are these fish aggressive within same-sex groups? Otherwise the conditions of husbandry look good.

The findings are very interesting, but can the authors shed some light on where the fish are going in the dyads? For example are female fish swimming around the male or are they swimming around the tank? This might be a useful discriminatory tool for explorations vs activity. It looks like there could be quite a lot of variance so could some of that be explained by females paying attention to the male rather than swimming away from the male, for example? I.e. is there explanatory power in ‘what’ females do as well as how far they go?

Author Response

REVIEWER 2

In section 2.1: The authors state that males and females were separated by sex; was water mixed between these tanks in a system, or were all tanks completely separate?

Response: All fish in Experiment 1 were housed in the same circulating water system. We added this information.

1st experiment: Was there any particular reason why water from the holding tanks was used? Is there a possibility that cues from the holding tank were therefore maintained within the experimental tank which could have influenced the result? For example are males likely to behave differently (more actively) in the presence of cues from male conspecifics? The authors do indicate that fish had experienced olfactory cues of other conspecifics so perhaps this isn't important. Presumably however holding tanks were separate (rather than in a system sharing water; see my first point) and therefore the water used in the experiment did not contain cues from the alternate sex?

Response: We used water from the holding tanks to minimize differences or disturbances caused by variation in the water parameters. So, for all trials, water used during the experimentation was the same the fish experienced in their holding tanks. While this water contained chemical cues from both sexes, we think that fish are behaving less stressed and exhibiting their normal behavioral repertoire when they do not experience any major difference in the water conditions.

One concern I have with the methodology is that activity as measured here could be conflated with exploration i.e. examining a new environment rather than moving around per se. Have the authors considered whether it is indeed activity they have measured? This may be particularly true since activity appears to be lower in the second trial (and in the discussion attributed to habituation). I think at the least within the methods the authors should reference other studies that have been conducted in this way and refer to the measure taken as ‘activity’. It is really important in behavioural science to ensure that the terminology used accurately reflects the behaviours being assessed, or at least that researchers are using the same terminology across studies. The authors do refer to both activity and exploration in the introduction.

Response: This is a valid point to think about. In animal behavior literature both terms are sometimes used to describe similar behavioral measures (e.g., the same trait). At the same time, some studies treat activity and exploration behavior as separate traits (see Reale et al. 2007 for a discussion). As exploration behavior includes moving around, which is activity, we do not believe that it can or must be seen as separate traits in our study. Of course, individuals may change their activity patterns when their environment becomes more familiar and we acknowledge this effect by referring to a habituation to our test environment. But is this a change from exploration to activity? We do not think this matters. Thus, whether we call it exploration or activity or both, it does not change anything in our interpretation of our data. Thus, we speak about both activity and exploration in the introduction and refer to broad-sense activity throughout.

On page 4 paragraph 3 the authors refer to ‘trail’ but I think this is a typo of ‘trial’?

Response: Thank you! We corrected this mistake.

What was the fate of the fish? Are these fish aggressive within same-sex groups? Otherwise the conditions of husbandry look good.

Response: After the experiments, all fish were transferred back into larger mixed-sex holding tanks and were not used for further experimentation (we put a statement on this at the first paragraph of the methods section). The guppy is assumed to be far less aggressive than other poeciliid species (see Bierbach et al. 2013 F1000research, Houde 1997, Magurran 2005) and we did not noticed overt aggressions in the holding tanks or during the experiments.

The findings are very interesting, but can the authors shed some light on where the fish are going in the dyads? For example are female fish swimming around the male or are they swimming around the tank? This might be a useful discriminatory tool for explorations vs activity. It looks like there could be quite a lot of variance so could some of that be explained by females paying attention to the male rather than swimming away from the male, for example? I.e. is there explanatory power in ‘what’ females do as well as how far they go?

Response: Again, this is a valid and interesting point. Nevertheless, we would like to point here to future studies that explicitly focus on tracking female behaviours in dyads as this may involve more sophisticated analysis software tools for automated tracking of non-stereotypic behaviors (like swimming paths comparisons or temporal correlations). This would definitely overstretch the current study. Also, more natural settings with shelter and hiding spaces could be used for that purpose, which in turn would unfortunately hamper proper tracking and automated detection of the involved animals.

Reviewer 3 Report

The manuscript “Male sexual preference for female swimming activity in the guppy (Poecilia reticulata)” deals with the interesting question how personality traits affect mate choice. In total the manuscript is thorough and well written.

I just have some clarification questions regarding the methods. Unfortunately, not all abbreviations are explained in the text (like MDM). Was the same camera and software used to in both experiments to analyze the behavior? Were the housing conditions in separate tanks between repeated measures the same for males and females? From the text it is not clear if the “pool of males” mentioned on page 4, 1. paragraph resembles the males tested before as figure 1 suggests. If this is the case, why were the males randomly allocated to the females and not intentionally paired depending on their activity pattern? It would have been easy to create pairs of low/low, low/high, high/low and high/high active males and females to test how they influence each other. Were new males used for the second experiment? The second sentence of the second paragraph, page 6 is misleading. The strength of preference (SOP) that you are calculating based on the times does not account for make size or females’ picture ID. Please clarify this.

Author Response

Responses to reviewer comments

REVIEWER 3

I just have some clarification questions regarding the methods. Unfortunately, not all abbreviations are explained in the text (like MDM). Was the same camera and software used to in both experiments to analyze the behavior? Were the housing conditions in separate tanks between repeated measures the same for males and females?

Response: Thank you very much for these comments. Due to the different experimental designs, we used different cameras in both experiments (model descriptions are now in the text also for experiment 1) but analysed the videos with the same software. For experiment one, males and females were housed similar between the repeated trials. Also, abbreviations are now explained in the text.

From the text it is not clear if the “pool of males” mentioned on page 4, 1. paragraph resembles the males tested before as figure 1 suggests.

Response: Yes, these were the same males. We changed to “pool of already tested males”.

If this is the case, why were the males randomly allocated to the females and not intentionally paired depending on their activity pattern? It would have been easy to create pairs of low/low, low/high, high/low and high/high active males and females to test how they influence each other. Were new males used for the second experiment?

Response: This is an interesting suggestion! However, as the analysis of the many videos took time, it was not possible to follow your suggested design. When pairs were created the experimenter did not know the swimming behavior of the involved fish yet. Nevertheless, a high/low pairing would also only aim at the edges of the swimming behavior continuum while our design involved fish from the whole range. In experiment 2, a new set of males was used (now mentioned in the text too).

The second sentence of the second paragraph, page 6 is misleading. The strength of preference (SOP) that you are calculating based on the times does not account for make size or females’ picture ID. Please clarify this.

Response: Thank you for this comment. We changed that sentence in order to make it clearer that the SOP score allowed for statistical analysis that included size and ids but did not included these variables itself.